# How Molecular Weight Cut-Offs and Physicochemical Properties of Polyether Sulfone Membranes Affect Peptide Migration and Selectivity during Electrodialysis with Filtration Membranes

**DOI:** 10.3390/membranes9110153

**Published:** 2019-11-13

**Authors:** Sabita Kadel, Geneviève Pellerin, Jacinthe Thibodeau, Véronique Perreault, Carole Lainé, Laurent Bazinet

**Affiliations:** 1Institute of Nutrition and Functional Foods (INAF), Dairy Science and Technology Research Centre and Department of Food Sciences, Université Laval, Québec, QC G1V 0A6, Canada; sabita.kadel.1@ulaval.ca (S.K.); genevieve.pellerin.3@ulaval.ca (G.P.); jacinthe.thibodeau.1@ulaval.ca (J.T.); veronique.perreault.5@ulaval.ca (V.P.); 2Laboratory of Food Processing and ElectroMembrane Processes (LTAPEM), Université Laval, Québec, QC G1V 0A6, Canada; 3Amer-sil, Kehlen 8281, Luxemburg; carole.laine@amer-sil.com

**Keywords:** electrodialysis with filtration membranes, membrane physicochemical properties, global peptide migration rate, peptide selectivity, molecular weight cut-offs

## Abstract

Filtration membranes (FMs) are an integral part of electrodialysis with filtration membranes (EDFM), a green and promising technology for bioactive peptide fractionation. Therefore, it is paramount to understand how physicochemical properties of FMs impact global and selective peptide migration to anionic (A^−^_RC_) and cationic (C^+^_RC_) peptide recovery compartments during their simultaneous separation by EDFM. In this context, six polyether sulfone (PES) membranes with molecular weight cut-offs (MWCO) of 5, 10, 20, 50, 100 and 300 kDa were characterized and used during EDFM to separate peptides from a complex whey protein hydrolysate. Surface charge, roughness, thickness and surface/pores nature of studied PES membranes were similar with small differences in conductivity, porosity and pore size distribution. Interestingly, global peptides migration to both recovery compartments increased linearly as a function of MWCO. However, peptide selectivity changed according to the recovery compartments and/or the peptide’s charge and MW with an increase in MWCO of FMs. Indeed, in A^−^_RC_, the relative abundance (RA) of peptides having low negative charge and MW (IDALNENK and VLVLDTDYK) decreased (45% to 19%) with an increase in MWCO, while the opposite for peptides having high negative charge and MW (TPEVDDEALEK, TPEVDDEALEKFDK & VYVEELKPTPEGDLEILLQK) (increased from 16% to 43%). Concurrently, in C^+^_RC_, regardless of MWCO used, the highest RA was observed for peptides having low positive charge and MW (IPAVFK & ALPMHIR). It was the first time that the significant impact of charge, MWCO and pore size distribution of PES membranes on a wide range of MWCO was demonstrated on EDFM performances.

## 1. Introduction

Membrane-based separation processes have been routinely used for the separation and purification of bioactive peptides obtained from different protein sources. In pressure-driven separation processes, the separation capacity of a filtration membrane (FM) is greatly affected by its pore size and pore size distribution (PSD) as they control either passage or rejection of a molecule through FM [1]. The pore size and PSD of commercially available ultrafiltration (UF) and nanofiltration (NF) membranes, the major types of membranes used for peptides separation/concentration in pressure-driven processes, are measured in terms of nominal molecular weight cut-offs (MWCO) by using solute rejection method [2]. MWCO is defined as the molecular weight of a test molecule (non-ionic solutes such as polyethylene glycol (PEG), oligostyrenes, alkanes and dextrans) that has a 90% rejection coefficient [1,2]. The effect of MWCO on the performance of UF/NF membranes in pressure-driven processes has been widely studied and is well understood [3,4]. In general, for a given material, an increase in MWCO, i.e., pore size of an UF/NF membrane results in a global increase in permeate flux or permeability and a global decrease in rejection coefficient, since it allows the passage of molecules with a wider range of molecular weights (MW). This consensus was supported by various studies, which were carried out for the separation of protein hydrolysate using ultrafiltration polyether sulfone (PES) [5] or cellulose acetate membranes [6]. However, the FMs with high permeability have been found to have low selectivity (separation of compound A in relation to compound B) and vice-versa [3]. Therefore, there is always a trade-off between permeability and selectivity given by a FM. Hence, MWCO of FM is one of the most important factors that affects the performances of pressure-driven processes.

However, the effect of MWCO of FM on the performance of electrodialysis with filtration membranes (EDFM), a promising technology for the separation and purification of bioactive peptides, has not yet systematically been investigated. Indeed, FMs are the integral part of EDFM, which are stacked in between ion exchange membranes. Therefore, separation of peptides is basically due to the combined effect of charge selectivity of electrodialysis and size exclusion capabilities of filtration membranes [7]. It is noteworthy that no transmembrane pressure is applied in the EDFM cell unlike pressure-driven separation processes and only the charged peptide having a charge migrates under the effect of the electric field, the neutral peptide theoretically stays in the hydrolysate solution and does not reach or pass the filtration membrane. In addition, EDFM does not use any chemical solvents during peptide fractionation and hence is considered as a green and eco-friendly technology. Furthermore, based on peptide selectivity, it is superior to pressure-driven processes since the peptides can be obtained in a purer form (as it can separate peptides having similar molecular mass but different charges, while pressure-driven processes cannot separate such peptides because the separation is solely based on the size of peptides) and avoids peptide fouling that is caused by the application of pressure, thereby, consequently improving its efficiency [8,9]. Therefore, EDFM stands as a potential alternative to pressure-driven processes for the separation of bioactive peptides or bioactive peptide fractions. Furthermore, this green technology has already been used successfully at lab/semi-pilot scale to selectively separate bioactive peptides from a wide range of complex hydrolysates [7]. Recently, EDFM allowed the separation of two bioactive peptide fractions from salmon protein hydrolysate, both increasing glucose uptake, decreasing hepatic glucose production and decreasing LPS-induced inflammation in macrophages and the identification of 4 new glucoregulatory peptide sequences [10]. Therefore, a clear understanding of the relation between MWCO and selective peptides separation, which affects the process performance, is crucial to select the appropriate MWCO for separating peptides of interest. 

In this context, six FMs with different MWCO, fabricated using the same membrane material, i.e., polyether sulfone (PES), and the same method, i.e., phase inversion, were tested in the present study in terms of global migration rate and selectivity of peptides obtained from the hydrolysis of whey protein isolate to EDFM recovery compartments. Therefore, the main objectives of this study were (1) to characterize the physicochemical properties of FMs to investigate if they exhibit different properties in spite of their same membrane material, (2) to study the global migration rate of peptides during this EDFM process as a function of MWCO/other membrane properties, (3) to identify the peptide sequences recovered in anionic and cationic recovery compartments during EDFM and (4) to study the peptide migration selectivity during EDFM as a function of MWCO/other membrane properties.

## 2. Materials and Methods

### 2.1. Chemicals

Acid and base such as HCl solution and NaOH, respectively, were bought from Fischer Scientific (Montréal, QC, Canada). Salts such as KCl and NaCl were obtained from BDH (VWR International Inc., Mississauga, ON, Canada). Na_2_SO_4_ was purchased from ACP Inc. (Montréal, QC, Canada).

### 2.2. Membranes

Commercial Neosepta cation-exchange membrane (CEM) and Neosepta anion-exchange membrane (AEM) were supplied by Astom (Tokyo, Japan). PES filtration membranes (FM) with MWCO of 5, 10, 20, 50, 100 and 300 kDa were obtained from Synder (Vacaville, CA, USA) (Commercial membrane code of these membranes are MT, ST, SM, MQ, LY and LX, respectively). All membranes used were made of the same material in order to emphasize the impact of the MWCO and physicochemical properties other than the membrane polymer on the performance of the EDFM process. The fact that the membrane polymer has as significant impact on the membrane’s performance was studied in [11]. In addition, the PES material was chosen since it is one of the most typically used material in the dairy industry [12]. Furthermore, previous studies on electro-membrane processes such as EDFM and electro-membrane filtration demonstrated that the MWCO of FM should be about 10 times higher than the size of proteins or peptides to obtain their successful transport through FM [13,14]. Otherwise, steric hindrance from the hydration layer slows down peptides migration. Therefore, in this study, we tested a wide range of MWCO to investigate the best MWCO required for the separation of a specific size range of peptides.

### 2.3. Whey Protein Hydrolysate

The whey protein hydrolysate, which was obtained by tryptic hydrolysis of a whey protein isolate (WPI), i.e., Prolacta 95 (Lactalis, Retiers, France) used in this experiment was the same as the one used previously by [15]. The identified peptides sequences along with their physicochemical characteristics are presented in Table 1. 

### 2.4. Electrodialysis Cell and EDFM Configuration

The electrodialysis (ED) cell (Model MP, ElectroCell AB, Sweden) and EDFM configuration used in this study were the same one as used by [15]. The effective surface area of ED cell was 100 cm^2^. The EDFM configuration consisted of one AEM, one CEM and two FMs: FM1 and FM2 with same MWCO placed in between AEM and CEM as shown in Figure 1. The configuration was comprised of four recirculation compartments: one compartment for anionic peptide recovery, one for cationic peptide recovery, one for feed solution and one for electrode rinsing solution (electrode rising loop was split into two circuits at the inlet of the cell). Peptide recovery compartments contained 2.5 L of KCl solution (2 g/L), feed compartment contained 2.5 L of 0.75% (*w*/*v*) whey protein hydrolysate, and electrode rinsing solution compartment contained 2.5 L of Na_2_SO_4_ solution (20 g/L). The electrode rinsing solution compartment was split into two streams going to the two electrolyte compartments of the cell. The solutions were circulated using four centrifugal pumps (CL3503, Baldor Electric Company, Fort Smith, AK, USA). Finally, the flow rates were monitored throughout the experiment using flow meters (F-550, Blue-White Industries Ltd., Huntington Beach, CA, USA).

### 2.5. Protocol

Before electroseparation, the physicochemical characteristics of all six FMs were analyzed: surface potential, roughness, thickness, contact angle, conductivity, porosity, nature of pores and pore size distribution.

Electroseparation of whey protein hydrolysate was carried out in a batch process with constant electric field strength of 2 V/cm. The experimental parameters such as pH, conductivity and temperature were controlled at 7,30 mS/cm and 20 °C, respectively throughout the experiment whatever the MWCO used, and that, according to previous experiments [15] to allow the comparison of all these results obtained in the same conditions. EDFM experiments were carried out for 120 min. Samples (2mL) were collected from the hydrolysate and each recovery compartments before applying voltage and at 30, 60, 90 and 120 min during EDFM to determine the migration kinetics of peptides over the experiment period. After EDFM, the recovered fractions from their respective compartments were freeze-dried and stored at 4 °C until analyzed for peptide concentration and sequence determination. Three independent EDFM experiments were carried out for each tested FMs. 

### 2.6. Analyses

#### 2.6.1. Membranes Physicochemical Properties

**a.** Surface Potential

Zeta potential was measured at room temperature using an electrokinetic analyzer (SurPASS, Anton Paar, Graz, Austria) equipped with a clamping cell with some modifications [16]. Briefly, the streaming current was measured in a 1 mM KCl solution in a pH range of 3 to 11. Three independent measurements were carried out for each membrane.

**b.** Roughness

For the measurement of surface roughness, a microstylus surface profilometer (DektakXT, Bruker Nano Surfaces, Tucson, AZ, USA) with 64-bit parallel processing operation was used which provides a 3-dimensional topography (image) of the flat membrane surface. The tip of the stylus with a radius of 12.5 μm was allowed to touch the membrane surface directly. The stylus traced across the membrane in hills and valley modes with vertical measurement range of 65.5 μm. Fifty separate scans with scan length and map extent of 1000 μm and scan duration of 5 s were carried out for each membrane. Vision64 (Bruker Nano Surfaces, Tucson, AZ, USA) was used as analysis software, which enables faster data crunching of large 3D map files as well as faster application of filters and multi scan database analyses. For each membrane, the roughness parameters: average roughness (Ra) and average peak-to-valley of the profile (Rz) were taken into consideration to evaluate the surface topography [15]. Measurements for each membrane were repeated three times on different coupons. 

**c.** Thickness

Marathon Electronic Digital Micrometer (Marathon Watch Company Ltd., Richmond Hill, ON, Canada) was used for the measurement of membrane thickness. The thickness was calculated by averaging six measurements at different locations on the membrane’s effective surface area.

**d.** Contact angle

Contact angle was measured by using a goniometer Theta OneAttension (Biolin Scientific, Linthicum Heights, MD, USA) and sessile drop method was used to determine the contact angle value [15]. Briefly, to remove excess water at the membrane surface, filter paper was used. Then, membranes were laid flat on a microscope slide and immobilized with small pieces of double-sided tape. A drop of distilled water (2 μL) was automatically dispensed on the membrane surface, then, a high-resolution camera captured the images. OneAttension software (Biolin Scientific, Espoo, Finland) was used for the analysis of drop profile, which generates constant contact angle data. The range of measurement was from 0° to 180°. This analysis was done in triplicate on the same membrane piece and repeated on three different pieces.

**e.** Conductivity

The conductivity of FM, AEM and CEM was measured using a specially designed clip from the Laboratoire des Matériaux Echangeurs d’Ions (Université Paris XII, Créteil, Val de Marne, France) as previously described by [17]. The conductivity was calculated by averaging six measurements at different locations on the membrane’s effective surface area.

**f.** Porosity, nature of pores and pore size distribution (PSD)

Porosity and nature of pores of FM was determined as described by [18]. However, an Automated Standard Porosimeter 3.2 (MPM&P Research Inc., Newmarket, ON, Canada) was used instead of Method of Standard Porosimeter with some modifications. Automated Standard Porosimeter is based on the laws of capillary equilibrium which measures the equilibrium distribution of working liquid between the sample and the standard (having known porometric curve). If pore size distribution of one of the porous bodies is known (i.e., standards), then the pore size distribution of other body (i.e., sample) can be calculated by determining the distribution of the liquid between all the porous bodies [18]. Automated standard porosimeter consists of sample holders, clamping device for making contact between the standards and the sample, drying devices, and an automatic manipulator for the assembling and disassembling of the stack of standards and sample, and for the transfer of the sample to the balance. Membranes were prepared in the shape of disc of 23 mm diameter. Octane and water were used as the working liquids to determine total and hydrophilic porosity (Octane has zero contact angle with almost all the materials), respectively. Vacuum impregnation of standards and test membranes except the membrane to be tested for hydrophilic porosity were done. Membrane to be tested for hydrophilic pores were soaked in water for at least 24 h at 4 °C after drying. POROVOZ software (MPM&P Research Inc., Toronto, ON, Canada) was used for the processing of measured data. Three independent experiments were carried out on three membrane coupons for both working liquids. 

As the membranes under our study were asymmetric ones (consisting of a filtrating layer-having small pores, and a support layer-having large pores), the porosity and pore size distribution (PSD) of the filtrating layer was also estimated. Normally, bi-porous substrate (asymmetric membrane in this case) exhibits two series of peaks/ types of peaks at the graph of differential distribution of pores volume (d*V*/d (log *r*)) vs pore radius (log *r*) [18]. Indeed, for asymmetric membranes, two series of peaks were observed: one series of peaks corresponding to the small log *r* values of the filtrating layer, and the second series of peaks to the large log *r* values of the support layer. The corresponding pore volume to the respective peaks is the porosity (cm^3^/cm^3^) value of filtrating and support layer. Additionally, PSD distribution of filtrating layer was calculated by dividing the corresponding volume of micropores (*r* ≤ 1nm), mesopores (1 < *r* ≤ 25 nm) and macropores (25 nm < *r*) by the porosity of the filtrating layer. 

#### 2.6.2. Total Peptide Concentration and Migration Rate in Recovery Compartments

Total peptide concentrations in anionic and cationic compartments were determined using micro BCA (µBCA) protein assay (Pierce, Rockford, IL, USA) from the samples withdrawn every 30 min over a period of 120 min. The absorbance was read at 562 nm on a microplate reader (xMark, Bio-Rad, Hercules, CA, USA). Concentration was determined with a standard curve in a range of 0-40 μg/mL of bovine serum albumin (BSA) [15].

For all FM, global peptide migration rate (g/m^2^·h) was calculated by dividing the total amount of peptides (g) migrated to anionic recovery compartment (A^−^_RC_)/cationic recovery compartment (C^+^_RC_) at the end of EDFM, determined by µBCA, by effective surface area of FM (m^2^) and duration of EDFM process (h). High concentration of peptides in recovery compartments is associated with its high migration rate in that compartment.

(1)$$Migration\;rate\;\left(g/m^{2}\cdot h\right)=\frac{Amount\;of\;peptide\;\left(g\right)}{Area\;\left(m^{2}\right)*Time\;\left(h\right)}$$

#### 2.6.3. Peptides Sequencing and Characterization

Peptides collected from the recovery compartments after EDFM were freeze-dried and thus obtained peptide powders were analyzed by ultra-high-performance liquid chromatography-mass spectrometry-quadrupole time-of-flight (UPLC-MS-QTOF) to identify their sequences by following the procedure explained by [19]. Briefly, same concentration, i.e., 0.5% (*w*/*v*) of freeze-dried peptides for all the samples were dissolved in 1 mL of UPLC grade water. Agilent Mass Hunter Software (Santa Clara, CA, USA) package was used for data acquisition and analysis (LC/MS Data Acquisition, Version B.08.00 and Qualitative Analysis for IM-MS, Version B.07.00 Service pack 2 with BioConfirm Software).

The relative abundance (RA) of each identified peptide was determined using the UPLC-UV chromatogram generated from the same sample injections that has described above. For each peptide, the area under the curve of each respective peak was divided by the sum of surface area of all the peaks detected for that sample and multiplied by 100 to obtain a percentage.

#### 2.6.4. Statistical Analyses

FM properties, peptide concentration and their migration rate to recovery compartments, and peptide relative abundance in recovery compartments were subjected to one-way analyses of variance (ANOVA). Student-Newman-Keuls (SNK) tests were carried out on data using SigmaPlot software (Version 12.0, Systat Software, San Jose, CA, USA) to determine which treatment was statistically different from others (significance level, *α* = 0.05). The global peptides migration to recovery compartments for each FM was compared by t-test (significance level, *α* = 0.05). Consequently, in the different statistical tests used, if the P-value is lower than the significance level of 0.05, then there is a difference between the treatment studied (FM properties, peptide concentration, migration rate …). 

All the measured values summarized in the table and represented in the graphs in the result section of this study are presented as mean value of three independent repetition ± standard deviation.

## 3. Results and Discussion

### 3.1. Physicochemical Properties of FM

All PES membranes (5, 10, 20, 50, 100 and 300 kDa) were characterized in terms of zeta potential, roughness, thickness, contact angle, conductivity, porosity, nature of pores and pore size distribution (Table 2). 

Globally, all FMs exhibited quite similar surface charge with an average zeta potential value of −12.25 ± 2.2 mV, which can be explained by the same material used for their fabrication [20], i.e., polyether sulfone as filtrating layer and polyester as support layer. However, the 10 kDa membrane would be slightly lower in zeta potential (~22% less) but still negatively charged and not significantly different from the 100 and 300 kDa. Similarly, all the FMs were found to have comparable or very close surface topography as demonstrated by their measured roughness values (average Ra = 0.98 ± 0.2 μm and Rz = 7.70 ± 2.2 μm). If no significant differences were observed for Ra, for roughness parameter Rz (probability level close to 5%), the 20 kDa membrane would be slightly higher than other membranes probably due to its high standard deviation. Concerning the values obtained for thickness, contact angle and hydrophilic porosity, they showed no trends (randomly increased and decreased with an increase in MWCO) (Table 2). Nonetheless, the surface of all FMs were slightly hydrophilic, having a contact angle value ranging between 59°–79° (if contact angle is less than 90°, the membrane is considered as hydrophilic, if higher than 90°, the membrane is hydrophobic [21]). All FMs had a high percentage (77–100%) of hydrophilic pores as indicated by hydrophilic porosity values (Table 2). In addition, the conductivity of FMs increased with an increase in their MWCO (Table 2). Similar results of increased conductivity with an increase in MWCO were reported by [22], when cellulose ester membranes of 10, 20, 50 and 100 kDa, conditioned in 0.1 M NaCl, were tested for their electrical conductivity. The authors suggested that the increased proportion of water volume due to a decrease in the volume occupied by the polymer must have caused an increase in membrane conductivity with an increase in MWCO. With an increase in FM conductivity, it lowers the resistance of the system and hence the energy consumption of the process [22]. Similarly, porosity and percentage of macropore distribution in the filtration layer showed direct relation with MWCO of FMs (increased with an increase in MWCO) (Table 2 and Figure 2). This suggested that the porosity and pore size of PES membranes increased with an increase in their MWCO.

It appears from these results that the physicochemical properties of PES membranes under this study were quite similar in terms of their surface charge, surface topography, thickness and nature (hydrophobicity/hydrophilicity) of surface and pores. However, these PES membranes were different according to their conductivity, porosity and macropore distribution values, which increased with an increase in MWCO of PES membranes. Furthermore, macropore distribution in the filtrating layer of such asymmetric membranes was directly related to the MWCO of PES membranes. 

In the following sections, the impact of these membrane properties on global and selective peptides migration to anionic and cationic recovery compartments will be discussed. 

### 3.2. Global Rate of Peptides Migration to Recovery Compartments

Global peptide migration to anionic (A^−^_RC_) and cationic (C^+^_RC_) recovery compartments as a function of time was found to be linear for all FMs and is presented in Figure 3a,b, respectively. Total peptide concentration in recovery compartments seemed to be dependent on the MWCO of FM: the highest migration rate was obtained for the membrane with the largest MWCO, i.e., 300 kDa (A^−^_RC_ = 11.28 ± 1.40 g/m^2^·h, C^+^_RC_ = 10.92 ± 1.65 g/m^2^·h), the lowest for the one having the smallest MWCO, i.e., 5 kDa (A^−^_RC_ = 2.41 ± 0.34 g/m^2^·h, C^+^_RC_ = 3.19 ± 0.15 g/m^2^·h) and intermediate values for the other membranes. However, for some low MWCO, whatever the peptide recovery compartment, no significant difference in terms of final global peptide migration was observed. Consequently, in A^−^_RC_, similar migration rates were observed for 5, 10 (3.11 ± 0.17 g/m^2^·h), 20 (3.17 ± 0.0.26 g/m^2^·h) and 50 kDa (3.19 ± 0.11 g/m^2^·h) but peptide migration through these membranes was significantly lower than 100 kDa (42–57% lower, *p* ≤ 0.004) and 300 kDa (72–79% lower, *p* < 0.001). Furthermore, the migration rate of anionic peptides through 100 kDa (5.50 ± 0.70 g/m^2^·h) was significantly lower (51% lower, *p* < 0.001) than 300 kDa. In C^+^_RC_, there was a more significant increase in peptide migration rate with an increase in MWCO except for 10 and 20 kDa. For these two membranes, similar rates of migration were obtained with an average value of 3.98 ± 0.13 g/m^2^·h, which was about 25% higher than 5 kDa (*p* = 0.033), 22% lower than 50 kDa (*p* = 0.019), 48% lower than 100 kDa (*p* < 0.001) and 64% lower than 300 kDa (*p* < 0.001). Furthermore, the peptide migration through 50 kDa (5.10 ± 0.32 g/m^2^·h) was 33% and 53% lower than 100 and 300 kDa, respectively (*p* < 0.001), and the migration rate for 100 kDa was 30% lower than 300 kDa (*p* = 0.001) with a value of 7.63 ± 1.37 g/m^2^·h. 

In addition, the global rate of peptide migration to A^−^_RC_ was significantly lower than to C^+^_RC_ (t test, *p* < 0.05) for all the membranes except for the ones having large MWCO, i.e., 100 and 300 kDa (Figure 3a,b). For these two membranes (100 and 300 kDa), no significant difference in the global rate of peptide migration to recovery compartment was noticed. These phenomena can be explained by the different types and degrees of interactions such as electrostatic and hydrophobic-hydrophilic, occurring between FM and peptides due to the zeta potential, contact angle and percentage of hydrophilic porosity values of studied FMs, respectively, as their MWCO increases. Zeta potential (ZP) is the measurement of apparent surface charge of FM, which determines the strength of electrostatic attraction or repulsion between FM and peptides resulting in either migration or rejection of peptides through the FM due to Donnan effect [9,23]. In our study, according to measured ZP value, all FMs were negatively charged. Therefore, electrostatic attraction must have occurred between membrane surface and positively charged peptides. This could have facilitated cationic peptides migration to C^+^_RC_ under the effect of electric field strength [9,23]. Indeed, it has been previously reported that global peptides migration through a membrane having opposite charge than that of peptide is much higher comparing to the ones having same charge during EDFM [15]. On the other hand, electrostatic repulsion between membrane surface and negatively charged peptides hindered the migration of anionic peptides to A^−^_RC_. Based on the measured values of contact angle and percentage of hydrophilic porosity, all FMs in our study were hydrophilic in nature. However, all identified cationic peptides were hydrophobic, while 75% of anionic peptides were hydrophilic (Table 1). Consequently, hydrophilic interactions must have occurred between membrane and anionic peptides because of their hydrophilic nature, which slowed down their migration through the hydrophilic membranes. On contrary, the migration of cationic peptides was not affected since repulsion occurs between membrane and peptides because of their different nature (hydrophilic and hydrophobic, respectively) [24]. Similar hydrophobic-hydrophilic interactions between membrane and peptides was reported by the recent study [15], which was carried out to separate the same protein hydrolysate as this study by EDFM. Therefore, electrostatic and hydrophilic-hydrophobic interactions (attraction/repulsion) between FM and peptides were found to strongly modify separation performance of filtration membranes during EDFM [15]. However, for 100 and 300 kDa, they had large MWCO and high percentage of macropores distributed in the filtration layer (100 kDa = 27% and 300 kDa = 88% vs 18% and lower for other membranes). When pore size of FM is much larger than the hydrodynamic size of molecules (see Table 1 for molecular mass of peptides), the molecules go through the membrane and do not come in contact with the membrane and the pores wall and, hence, no kind of interactions (electrostatic/hydrophobic-hydrophilic) occurs/or those interactions are largely reduced between FM and peptides [16,25,26,27]. Therefore, these results indicated that the migration of charged peptides through charged membranes is governed by electrostatic/hydrophobic-hydrophilic interactions between FM and peptide up to or only for a certain range of MWCO of FM. For FMs having large MWCO, the effect of MWCO (size exclusion) is more prominent rather than their charge and nature, when global peptide migration rate is considered. This result is in agreement with the results obtained by [15], where similar rates of global peptide migration to both recovery compartments were obtained when negatively charged and large pore size FMs; PVDF 800 kDa and CF55 (average pore radius = 0.26μm) were used during EDFM to separate the same hydrolysate as used in this study. 

When the graph was plotted between MWCO and the global peptide migration rate, for both recovery compartments (A^−^_RC_ (Figure 4a) and C^+^_RC_ (Figure 4b)), a linear relationship was observed between these two parameters, even though similar rates of peptides migration was noticed for few low MWCOs (A^−^_RC_ = 5, 10, 20 and 50 kDa; C^+^_RC_ = 10 and 20 kDa) (Figure 4). Such relation could be explained by relating MWCO of FM with macropore distribution on its filtrating layer. For studied FMs, their MWCO have been found to have linear relation with macropore distribution (pore size > 50 nm) in their filtrating layer (Figure 2). The lowest value for macropore distribution was observed for 5 kDa (5 ± 1%), the highest for 300 kDa (88 ± 2%) and with intermediate values for the other studied membranes. The increase in macropore differential volume would mean an increase in average pore size of FM, which apparently allows peptides with a wide range of molecular mass to migrate through the membrane. In addition, as mentioned before, conductivity and porosity of these FMs increased as MWCO increases, which is associated with high migration/permeate flux due to decreased global resistance of EDFM system [28] and high membrane permeability in pressure-driven processes [29], respectively. Indeed, the permeate flux in pressure-driven processes is proportional to the pore size and porosity of FM according to the general rule of membrane processes [30,31]. Similar phenomena could have happened in EDFM as well. Therefore, the combined effect of macropore distribution in filtrating layer, porosity and conductivity of FM would have contributed to linear increase in total peptide migration rate as a function of MWCO in A^−^_RC_ (R^2^ = 0.9871) and C^+^_RC_ (R^2^ = 0.9418) (Figure 4a,b).

### 3.3. Individual Peptide Migration to Recovery Compartments: Peptides Selectivity

The major peptides recovered from A^−^_RC_ and C^+^_RC_ were identified and listed with their relative abundance (in terms of area under the peak of UV chromatogram) in Table 3 and Table 4, respectively. As expected, regardless of MWCO of FMs used, all negatively charged peptides (8 peptides, 7 peaks) present in the initial hydrolysate migrated to A^−^_RC_ and all positively charged peptides (5 peptides, 5 peaks) to C^+^_RC_, suggesting the selective separation based on their charge (see physicochemical properties of peptides in Table 1). However, the relative abundance (relative abundance of a specific peptide is proportional to its concentration) of each peptide in respective compartments varied based on MWCO of used FM. At the same time, two out of three neutral peptides, i.e., GLDIQK and VAGTWY were observed in both recovery compartments.

#### 3.3.1. Selective peptide migration to A^−^_RC_

In A^−^_RC_, based on their molecular weight (MW) and charge, anionic peptides could be classified into three groups: AP1, AP2 and AP3. AP1 includes peptides having low MW and negative charge: IDALNENK (MW = 915.46 Da, Charge = −1) and VLVLDTDYK (MW = 1064.57 Da, Charge = −1), AP2 includes peptides having high MW and negative charge: TPEVDDEALEK (MW = 1244.58 Da, Charge = −4), TPEVDDEALEKFDK (MW = 1634.76 Da, Charge = −4) and VYVEELKPTPEGDLEILLQK (MW = 2312.25 Da, Charge = −3), and AP3 includes peptides having high MW and low negative charge: SLAMAASDISLLDAQSAPLR (MW = 2029.05 Da, Charge = −1) and WENGECAQK + LSFNPTQLEEQCHI (MW = 2719.19 Da, Charge = −1, −2). 

The highest RAs of peptides belonging to AP1 group was obtained when PES membrane with MWCO of 5 kDa was used, representing about 45% of the total peptide abundance. Surprisingly, their RAs decreased with an increase in MWCO of PES membranes. Consequently, their lowest RAs was noticed when 300 kDa membrane, the highest MWCO used in this study, was used (representing about 19% of the total peptide abundance). On the other hand, the result was just opposite for AP2 group of peptides; their lowest RAs was observed when 5 kDa PES membrane was used, representing about 16% of total peptide abundance. The RAs of this group of peptides was increased with an increase in MWCO of PES membranes with 300 kDa membrane giving their highest RAs, representing about 43% of the total peptide abundance (Table 3). Regarding AP3 group of peptides, their RAs were more or less similar regardless of MWCO used, representing about 13% of total peptide abundance for all FMs (Table 3). Such performances can be explained by the varying degrees effects of steric hindrance (due to MWCO of FM) and electrostatic repulsion (due to negative surface charge of FM) on different group of peptides. However, the effect of hydrophilic interactions would be more or less similar for all groups of peptides for a specific MWCO (if peptide comes in contact with FM) since all FMs and peptides (except VLVLDTDYK and SLAMAASDISLLDAQSAPLR) in this study were hydrophilic. The effect of steric hindrance and electrostatic repulsion to AP1 group of peptides at membrane interface was reduced/negligible because of their low MW and charge, respectively, comparing to other group of peptides. Therefore, their migration through 5 kDa membrane was not hindered due to these effects. Similar results relating to the effects of steric hindrance and electrostatic repulsion was observed during the separation of bioactive peptides from whey protein hydrolysate using EDFM [15] and α_s2_,-casein f (183–207) from α_s2_,-casein hydrolysate using Electro-membrane filtration (EMF) with 20 kDa membrane [13]. On the contrary, the effects of electrostatic repulsion and the steric hindrance were important for AP2 group of peptides due to their high MW and high negative charge, when FM having low MWCO such as 5 kDa was used. Consequently, their migration through such FM was slowed down. This result is in agreement with previous result obtained by [25] during electrophoretic membrane contractor (EMC), which showed the combined effect of charge and size of beta-lactoglobulin (MW = 18.3 kDa, Isoelectric point = 4.4) on its separation by 30 kDa cellulose acetate membrane (a negatively charged membrane) at pH 8. At the same time, it is important to note that as MWCO increases, the steric effects of FM on peptides migration decrease due to an increase in effective pore size [13,16] and macropore distribution in the filtrating layer of FM (Table 2). In addition, the electrostatic interactions between membrane surface and charged peptides also reduced as MWCO increases [25]. Therefore, the rate of peptide migration would be predominantly dependent on their own physicochemical properties, i.e., molecular weight and charge, rather than on FM properties for a same PES membrane material. Furthermore, studies have shown that, under the effect of electric field during electrophoresis, the migration rate of a peptide was directly related to its charge and inversely to its molecular mass [32]. Additionally, in general, charge of peptides/molecules plays a more important role than their MW in their migration through charged membranes under the electric field [16,32]. In this study, all peptides belonging to AP2 group were highly negatively charged, while AP1 group were weakly negatively charged. This may have facilitated the migration of AP2 group of peptides comparing to AP1 group of peptides given that the steric effects and electrostatic repulsions at FM interface were reduced/negligible. This explains why there was an increase in RA of AP2 group of peptides with an increase in MWCO of FMs. Regarding AP3 group of peptides, their migration was slowed down due to their low charge and high MW. Indeed, the ratio mass/charge is the most important parameter.

The decreased RA of AP1 group of peptides does not mean that their migration rate decreased as MWCO increases, but, in reality, was due to the increased migration rate of AP2 group of peptides as MWCO and competitive migration effects increase [33]. In other words, the increase in migration rate of AP2 group of peptides with an increase in MWCO of PES membranes was much higher compared to the migration rate of AP1 group of peptides due to their low charge. Indeed, since the same total concentrations of peptides (0.5%, *w*/*v*) solution for all the samples were analyzed by UPLC-MS for abundance evaluation, an increase in RA of AP2 group of peptides corresponding to its higher concentration, was linked with a decrease in concentration of AP1 group of peptides resulting to its lower RA.

#### 3.3.2. Selective Peptide Migration to C^+^_RC_

Like in A^−^_RC_, cationic peptides that migrated to C^+^_RC_ were classified into three major groups: CP1, CP2 and CP3. CP1 includes peptides having MW ranging between 650-850 Da and low positive charge (+1): IPAVFK (MW = 673.42 Da) and ALPMHIR (MW = 836.47 Da), CP2 includes peptides with MW between 900-1200 Da and low positive charge (+1): LIVTQTMK (MW = 932.53 Da), and VGINWLAHK (MW = 1199.65) and CP3 includes only TKIPAVFK having a MW of 902.56 Da and a charge of +2. In this compartment, the RA of these three groups of peptides (CP1, CP2 and CP3) obtained by using different FMs showed similar trend as that of AP1, AP2 and AP3 groups of peptides in A^−^_RC_ for the same FMs, respectively: RA of peptides belonging to CP1 group was decreased with an increase in MWCO, CP2 group was increased with an increase in MWCO and CP3 group was similar regardless of MWCO used (Table 4). However, the increase or decrease in RA of a specific peptide with an increase in MWCO in C^+^_RC_ was not as noticeable as in A^−^_RC_. Consequently, regardless of any MWCOs used, the highest RA in C^+^_RC_ was given by peptides belonging to CP1 group (Table 4), which represented 47-62% of total peptide abundance for FMs with MWCO ranging from 5 to 300 kDa. This could be explained by the low charge (+1 and +2) and narrow-range MW (673–1200 Da) of cationic peptides unlike anionic peptides whose charge and MW were wide-ranging (Charge = −1, −3, −4; MW = 915–2720 Da). Therefore, the electrostatic attraction (since peptides and membranes were oppositely charged), which facilitates the transport of positively charged peptides through negatively charged FM under the effect of electric field, must be quite similar for all cationic peptides for a specific FM. At the same time, there was a negligible effect of steric hindrance to CP1 group of peptides due to their low MW and only a very little effect on CP2 and CP3 group of peptides even when FM with the smallest MWCO, i.e., 5 kDa was used. Also, the effect of hydrophobic-hydrophilic interactions would be the same since all FMs were hydrophilic (see the values of contact angle and hydrophilic porosity in Table 2) and cationic peptides were hydrophobic (see GRAVY score of peptides in Table 1). Moreover, as stated before, the effect of electrostatic and steric hindrance became negligible as MWCO increases [13,16,25], and the migration of peptides through FM would be mainly dependent on their charge and molecular weight under the effect of electric field [32]. Additionally, RA of a specific peptide in the feed solution (initial hydrolysate) also affects its migration through the FM, i.e., higher the RA in initial hydrolysate, higher is its migration through FM [14], given that the effect of steric hindrance and electrostatic interactions are negligible. 

As mentioned previously for peptides recovered in A^−^_RC_, the slight decrease in RAs of CP1 group of peptides with an increase in MWCO was mainly due to the increase in RAs of CP2 group of peptides. However, it doesn’t mean that the migration rate of CP1 group of peptides decreased with an increase in MWCO, it just means that the transport rate of CP2 group of peptides increased with an increase in MWCO due to reduced/negligible steric effect and high competitive migration of CP2 group of peptides, consequently reducing the RAs of CP1 group of peptides [33]. 

Unexpectedly, the negatively charged peptide SLAMAASDISLLDAQSAPLR (MW = 2029, Charge = −1) also migrated through 100 and 300 kDa, but in a very low quantity (RA = 2.3–3.6%). This peptide may have aggregated with other cationic peptides (increasing consequently its charge) via peptide-peptide interactions such as hydrophobic interactions and migrated through FMs (100 and 300 kDa), since steric hindrance correlated with the high MWCO of these FM is low. Indeed, SLAMAASDISLLDAQSAPLR and all the cationic peptides identified/present in the whey protein hydrolysate used in this study had positive GRAVY score (meaning that they are globally hydrophobic) (see Table 1 for GRAVY score of peptides). Indeed, it has been previously demonstrated that most of the peptides obtained by the tryptic hydrolysis of β-lactoglobulin (all the identified peptides obtained by the tryptic hydrolysis of WPI in this study were coming from β-lactoglobulin) were participating in hydrophobic interactions [34,35]. The same phenomena (hydrophobic interactions) may explain the migration of negatively charged VLVLDTDYK to C^+^_RC_ through all the membranes, which co-eluted with positively charged peptide VGINYWLAHK in UPLC-UV chromatogram. 

Surprisingly, two out of three neutral peptides present in the initial WPH, i.e., GLDIQK (672.3801 Da) and VAGTWY (695.3261 Da) were observed in both recovery compartments. They may have migrated along with charged peptides due to peptide-peptide interactions (hydrophobic-hydrophilic interaction), maybe due to the electroosmotic flow, but no information was available in the literature, or diffusion as observed in pressure-driven processes by [34]. In A^−^_RC_, their RA for different FMs followed the similar trend as that of AP1 group of peptides: their RA decreased with an increase in MWCO. However, this doesn’t mean that their migration rate decreased with an increase in MWCO of FMs. As explained earlier for AP1 group of peptides, the lower RA of neutral peptides was due to high and competitive migration rate of AP2 group of peptides (due to their high charge) as MWCO of FMs increases. On the other hand, their RA in C^+^_RC_ was comparable whatever the MWCO was used. This is because all the cationic peptides present in the hydrolysate in this study were small (MW < 1200 Da) and slightly positively charged, and therefore, their migration rate through any FMs were not affected by steric hindrance/electrostatic repulsion. This resulted in more or less similar RA of a specific cationic peptide for all the FMs and consequently to comparable RA of neutral peptides for any MWCO tested. 

## 4. Conclusions

MWCO of PES membranes used during EDFM has a major impact on the global peptide migration to both recovery compartments. Indeed, since the physicochemical properties of the tested FMs were alike for all samples; the global/total peptide migration linearly increased with an increase in MWCO of PES membranes. However, the selectivity in individual peptide migration to their respective recovery compartments changed as the MWCO increased and this selectivity was different according to the recovery compartment or the peptide’s physicochemical characteristics. The effect of MWCO was more noticeable for anionic peptide selectivity than for cationic peptide selectivity. Hence, in A^−^_RC_, the selectivity of low molecular weight (<1065 Da) and slightly negatively charged (−1) peptides was more important for FM having low MWCO: RA of these peptides decreased with an increase in MWCO of FMs. On the contrary, the trend was just opposite for high molecular weight (1244–2313 Da) and highly negatively charged (−3 or −4) peptides: their RA increased with an increase in MWCO of FMs. These tendencies were explained by the different degrees of steric effect and electrostatic repulsion occurring between low MWCO FM-low MW peptides, and large MWCO FM-large MW peptides. Concerning peptides recovered in C^+^_RC_, the selectivity of low MW and weakly positively charged peptides was important regardless of MWCO used, though a slight decrease in their RA was noticed with an increase in MWCO. Based on these results, it can be concluded that the MWCO have a significant impact on the selective separation of WPI hydrolysate anionic peptides but little effect on cationic peptides when negatively charged FMs (PES) were used. At the same time, it is important to note that the MW and charge of peptides also played an important role in their transport phenomena through FMs during EDFM. Consequently, for peptide selectivity, the MWCO, the charge and the macropore distribution of FMs as well as the peptides characteristics, mainly the mass and the charge, must be considered to optimize their individual separation.

## Figures and Tables

**Figure 1 membranes-09-00153-f001:**
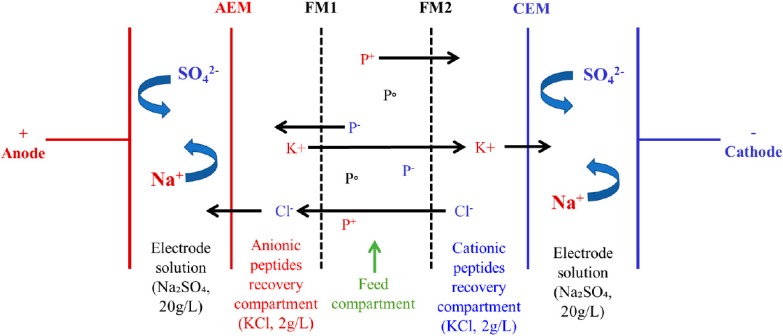
EDFM cell configuration for the simultaneous separation of anionic and cationic whey protein hydrolysate. FM1 and FM2 are filtration membranes with same MWCO, AEM: anion-exchange membrane and CEM: cation-exchange membrane, P^+^: cationic peptide, P^−^: anionic peptide, P^0^: neutral peptide (adapted from [15]).

**Figure 2 membranes-09-00153-f002:**
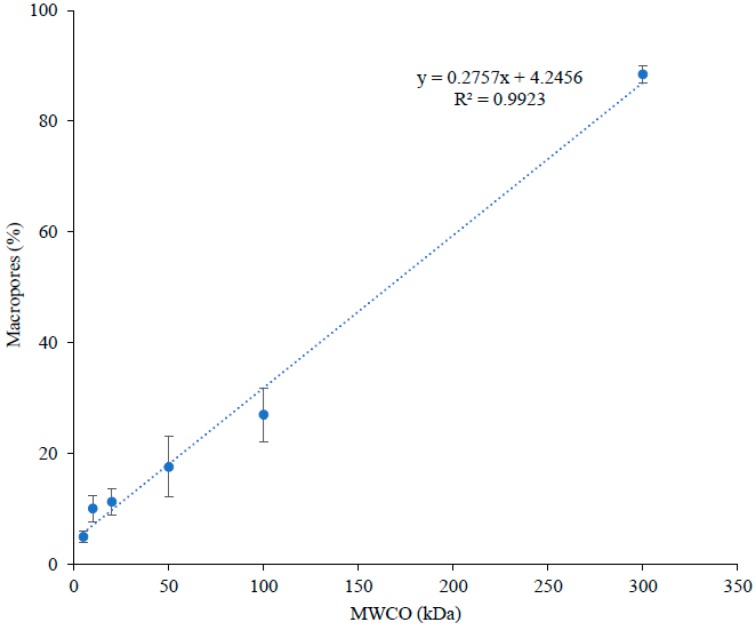
Distribution of percentage macropores as a function of MWCO.

**Figure 3 membranes-09-00153-f003:**
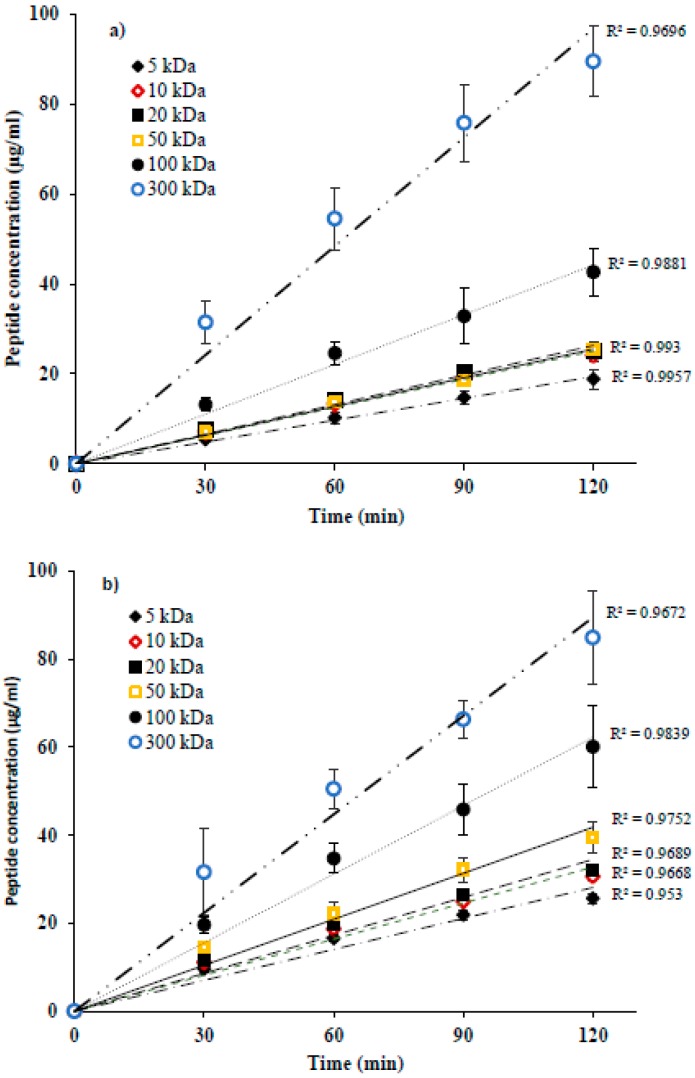
Evolution of peptides concentrations in (**a**) anionic and (**b**) cationic peptides recovery compartments as a function of time during EDFM.

**Figure 4 membranes-09-00153-f004:**
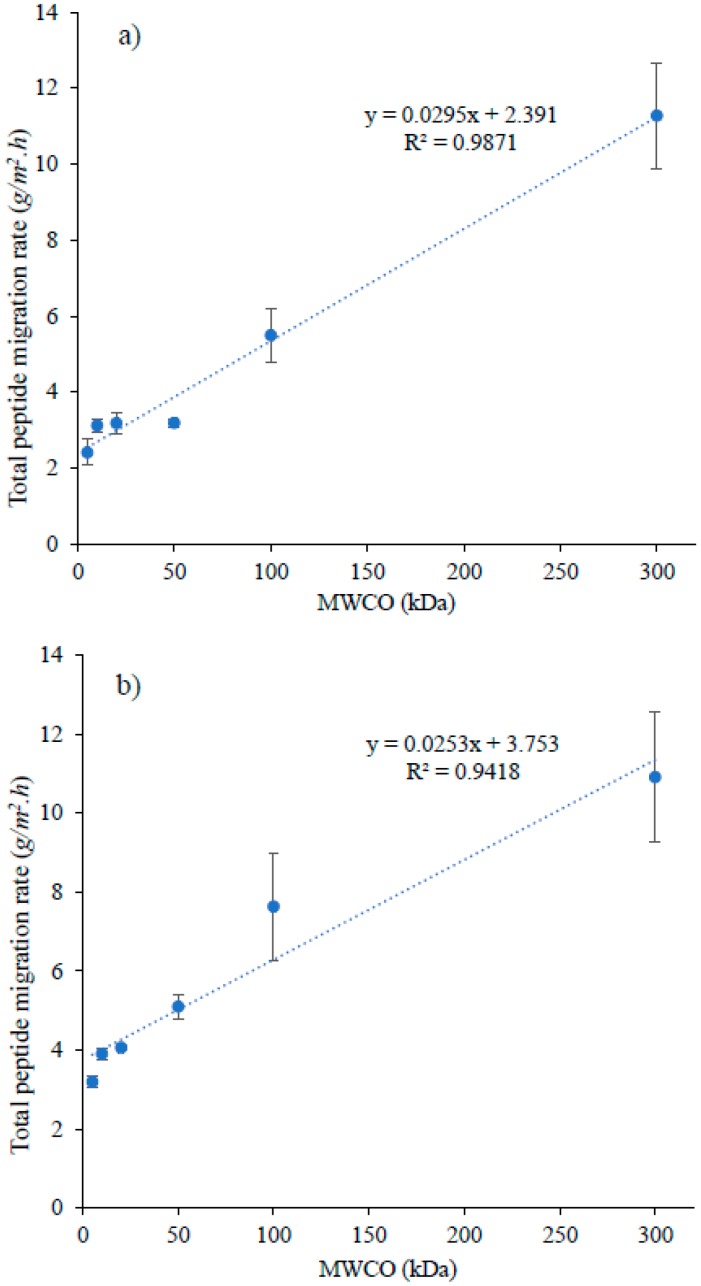
Rates of total peptide migration (g/m^2^·h) to (**a**) anionic and (**b**) cationic recovery compartment as a function of MWCO.

**Table 1 membranes-09-00153-t001:** Characteristics of major peptides obtained after tryptic hydrolysis of a whey protein isolate [15].

Peptide Sequences	Observed Molecular Mass (Da)	Global Charge at pH 7	GRAVY	Relative Abundance	Peptide Source
GLDIQK	672.38	0	−0.5	3.5 ± 0.2	BLG
IDALNENK	915.46	−1	−1	4.7 ± 0.2	BLG
ALPMHIR	836.47	1	0.386	6.1 ± 0.2	BLG
LIVTQTMK	932.53	1	0.7	4.5 ± 0.2	BLG
TKIPAVFK	902.56	2	0.4	1.4 ±0.5	BLG
TPEVDDEALEK	1244.58	−4	−1.264	5.5 ± 0.3	BLG
VLVLDTDYKK	1192.67	0	−0.1	8.1 ± 0.7	BLG
IPAVFK	673.42	1	1.3		BLG
TPEVDDEALEKFDK	1634.76	−4	−1.321	2.1 ± 0.1	BLG
VAGTWY	695.33	0	0.5	12.4 ± 0.3	BLG
VLVLDTDYK	1064.57	−1	0.344	10.2 ± 0.1	BLG
VGINYWLAHK	1199.65	1	0.11		ALA
Not identified	2905.30	−	-	2.1 ± 0.2	-
WENGECAQK + LSFNPTQLEEQCHI	2719.19	−1 −2	−1.7 −0.1	3.6 ± 0.4	BLG
Not identified	1262.66	-	-		-
Not identified	2777.20	-	-	4.5 ± 0.1	-
VYVEELKPTPEGDLEILLQK	2312.25	−3	−0.3	15.1 ± 0.3	BLG
SLAMAASDISLLDAQSAPLR	2029.05	−1	0.54	11.8 ± 0.1	BLG
Not identified	3313.53	-	-	4.2 ± 0.1	-

GRAVY: Grand Average of Hydropathy index. It qualifies and quantifies the hydrophobic or hydrophilic character of each peptide. A positive GRAVY score is related to globally hydrophobic peptide, whereas a negative GRAVY score to globally hydrophilic peptide. BLG: Beta-lactoglobulin, ALA: Alpha-lactalbumin

**Table 2 membranes-09-00153-t002:** Physicochemical properties of filtration membranes measured prior EDFM.

	5 kDa	10 kDa	20 kDa	50 kDa	100 kDa	300 kDa	*P*-Value
**Zeta Potential (mV)**	−14.7 ± 1.6^a^	−9.6 ± 0.6^b^	−13.6 ± 2.3^a^	−13.4 ± 0.8^a^	−11 ± 1.2^ab^	−11.2 ± 1.6^ab^	*p* ≤ 0.038
**Roughness (Ra)**	1.0 ± 0.3^a^	0.8 ± 0.2^a^	1.2 ± 0.10^a^	1.0 ± 0.2^a^	0.8 ± 0.0^a^	1.2 ± 0.2^a^	*p* < 0.05
**Roughness (Rz)**	7.4 ± 1.5^b^	5.5 ± 1.4^bc^	10.3 ± 2.4^a^	9.0 ± 2.1^b^	5.9 ± 0.5^ab^	8.1 ± 1.4^b^	*p* ≤ 0.045
**Thickness (μm)**	186 ± 5^ab^	179 ± 12^a^	186 ± 8^ab^	203 ± 4^b^	187 ± 6^ab^	221 ± 4^c^	*p* ≤ 0.010
**Contact Angle (°)**	65 ± 3^a^	59 ± 2^a^	72 ± 6^b^	79 ± 6^c^	70 ± 1^b^	62 ± 3^a^	*p* ≤ 0.041
**Hydrophilic Porosity (%)**	77 ± 2^a^	100 ± 0^c^	100±0^c^	84 ± 4^b^	85 ± 4^ab^	89 ± 4^b^	*p* ≤ 0.050
**Conductivity (mS/cm)**	4.2 ± 0.2^a^	4.8 ± 0.3^b^	5.0 ± 0.2^b^	6.4 ± 0.2^d^	5.7 ± 0.3^c^	9.1 ± 0.5^e^	*p* ≤ 0.023
**Total Porosity (cm^3^/cm^3^)**	0.43 ± 0.02^a^	0.42 ± 0.02^a^	0.47 ± 0.02^ab^	0.52 ± 0.05^b^	0.52 ± 0.00^c^	0.60 ± 0.01^c^	*p* ≤ 0.019
**Porosity of filtration layer (cm^3^/cm^3^)**	0.20 ± 0.01^ab^	0.17 ± 0.01^a^	0.22 ± 0.03^abc^	0.25 ± 0.05^bc^	0.27 ± 0.04^c^	0.44 ± 0.02^d^	*p* ≤ 0.028
**Macropores in filtration layer (%)**	5 ± 1^a^	10 ± 2^a^	11 ± 2^a^	18 ± 5^b^	27 ± 5^c^	88 ± 2^d^	*p* ≤ 0.046

Values in the same row followed by the same letter in the superscript are not significantly different (One-way ANOVA, SNK test, *P* > 0.05). *P*-value column presents the ANOVA probability level for each respective FM property in terms of significant differences between all FMs studied.

**Table 3 membranes-09-00153-t003:** Relative abundance of identified peptides recovered from anionic recovery compartment after 120 min of EDFM for PES membranes with MWCO of 5, 10, 20, 50, 100 and 300 kDa.

Peptides (Grouping)	Relative Abundance (%)					
	5 kDa	10 kDa	20 kDa	50 kDa	100 kDa	300 kDa
IDALNENK (AP1)	18.9 ± 3.0^d^	16.0 ± 1.1^c^	14.4 ± 1.9^c^	11.4 ± 0.4^b^	10.2 ± 0.3^ab^	7.9 ± 0.7^a^
VLVLDTDYK (AP1)	25.7 ± 1.4^d^	24.7 ± 0.6^d^	11.3 ± 2.2^a^	16.3 ± 1.0^c^	13.7 ± 0.4^b^	11.0 ± 0.2^a^
TPEVDDEALEK (AP2)	6.6 ± 2.4^a^	9.0 ± 0.8^a^	14.2 ± 0.1^bc^	13.3 ± 1.2^b^	16.4 ± 0.6^cd^	17.3 ± 1.8^d^
TPEVDDEALEKFDK (AP2)	2.0 ± 1.2^a^	2.1 ± 0.4^a^	4.0 ± 0.9^b^	2.5 ± 0.4^a^	5.0 ± 0.1^b^	5.3 ± 0.6^b^
VYVEELKPTPEGDLEILLQK (AP2)	7.8 ± 3.5^a^	10.4 ± 0.7^a^	14.1 ± 1.2^b^	16.1 ± 1.5^b^	17.3 ± 0.3^b^	20.5 ± 1.0^c^
SLAMAASDISLLDAQSAPLR (AP3)	7.3 ± 2.7^a^	12.1 ± 2.1^a^	8.4 ± 2.3^a^	8.5 ± 1.7^a^	8.3 ± 1.0^a^	9.4 ± 0.9^a^
WENGECVAQK+ LSFNPTQLEEQCHI/Not identified (AP3)	3.2 ± 0.3^a^	4.7 ± 0.5^b^	5.2 ± 0.4^b^	4.7 ± 0.6^b^	4.6 ± 0.8^b^	4.4 ± 0.5^b^
VAGTWY	19.8 ± 6.9^c^	10.6 ± 1.1^ab^	13.2 ± 3.2^bc^	10.7 ± 1.8^ab^	9.0 ± 0.8^ab^	7.7 ± 0.9^a^
GLDIQK	5.4 ± 1.6^b^	4.0 ± 0.6^ab^	3.7 ± 0.8^ab^	3.4 ± 0.5^a^	3.0 ± 0.1^a^	2.5 ± 0.5^a^
Not identified (MW = 2905.30 Da)	0^a^	2.5 ± 0.3^b^	2.8 ± 0.4^b^	2.6 ± 0.4^b^	2.7 ± 0.0^b^	2.8 ± 0.4^b^
Not identified (MW = 2777.20 Da)	3.3 ± 0.5^a^	3.9 ± 0.8^a^	5.4 ± 0.4^bc^	6.4 ± 0.9^cd^	6.1 ± 0.3^cd^	7.1 ± 0.6^d^
Not identified (MW = 3313.53 Da)	0^a^	3.0 ± 0.7^b^	3.2 ± 0.0^bc^	4.1 ± 0.4^bc^	3.6 ± 0.1^c^	4.1 ± 0.3^c^

Values in the same row followed by the same letter in the superscript are not significantly different (One-way ANOVA, SNK test, *P* > 0.05).

**Table 4 membranes-09-00153-t004:** Relative abundance of identified peptides recovered from cationic recovery compartment after 120 min of EDFM for PES membranes with MWCO of 5, 10, 20, 50, 100 and 300 kDa.

Peptides (Grouping)	Relative Abundance (%)					
	5 kDa	10 kDa	20 kDa	50 kDa	100 kDa	300 kDa
IPAVFK (CP1)	32.6 ± 0.2^c^	33.2 ± 2.2^c^	30.2 ± 0.4^bc^	32.5 ± 2.1^c^	27.8 ± 1.2^ab^	25.2 ± 0.5^a^
ALPMHIR (CP1)	28.9 ± 2.4^b^	26.1 ± 2.8^ab^	25.9 ± 0.5^ab^	23.0 ± 2.8^a^	23.4 ± 0.9^a^	21.2 ± 1.1^a^
LIVTQTMK (CP2)	2.9 ± 0.7^a^	5.0 ± 2.0^ab^	6.0 ± 1.6^bc^	6.7 ± 0.5^bc^	6.8 ± 1.0^bc^	8.7 ± 0.7^c^
VLVLDTDYK/ VGINYWLAHK (CP2)	3.6 ± 0.5^a^	4.4 ± 1.4^ab^	6.0 ± 0.5^bc^	5.5 ± 0.9^b^	5.9 ± 0.4^bc^	7.4 ± 0.1^c^
TKIPAVFK (CP3)	10.1 ± 0.2^a^	9.8 ± 2.2^a^	8.9 ± 0.6^a^	7.5 ± 0.7^a^	8.4 ± 0.4^a^	7.6 ± 0.7^a^
GLDIQK	4.9 ± 0.1^a^	5.2 ± 0.3^ab^	5.3 ± 0.4^ab^	5.2 ± 0.2^ab^	5.9 ± 0.3^c^	5.5 ± 0.1^bc^
VAGTWY	16.9 ± 1.9^a^	16.4 ± 2.4^a^	17.9 ± 0.7^a^	18.2 ± 1.2^a^	18.4 ± 0.2^a^	19.0 ± 0.1^a^
IDALNENK	0	0	0	1.3 ± 0.2^a^	1.1 ± 0.3^a^	1.8 ± 0.4^a^
SLAMAASDISLLDAQSAPLR	0	0	0	0	2.3 ± 0.7^a^	3.5 ± 0.3^b^

Values in the same row followed by the same letter in the superscript are not significantly different (One-way ANOVA, SNK test, *P* > 0.05).

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
