# Peer review of "How Molecular Weight Cut-Offs and Physicochemical Properties of Polyether Sulfone Membranes Affect Peptide Migration and Selectivity during Electrodialysis with Filtration Membranes"

_membranes, 2019, doi:10.3390/membranes9110153_

Round 1

Reviewer 1 Report

The article covers the comparison of different UF membranes for use in separation of a peptide mixture with EDFM technology. The work is very interesting, the experimental approach is of high quality and the paper is generally well written. Some improvements are required however.

General remarks:

Statistical tests should be reported in scientific format to avoid any doubt. Please check:

Table 2: Anova tests (degrees of freedom, F-statistic and p-value) e.g. F(1, 24) = 44.4, p < .001.

Table 2,3,4: Indicate somewhere what x ± y refers to,  is it mean and SD (not SEM? N?)

Table 2: Clarify what the column P-value refers to.

L245: Explain very briefly (1-2 sentences) what these stat tests do (null hypothesis, assumptions/levene’s test, what result looks like, how result is interpreted) to improve readability, as you use them a lot in the results section.

L250: significance level, alpha = 0.05.

L317: “ the global rate of peptide migration to A-RC was significantly lower than to C+RC (P<0.05) “ which test is performed here?

Section 3.3: when quantitatively comparing RA slopes of different peptides groups (AP1-AP3, CP1-CP3) among each other, ancova or regression analysis should be used. Otherwise it should be mentioned specifically on which basis qualitative comparison is made.

Section 3.1 should be improved, not all information required to follow the story is presented in a clear way.

Table 3 and 4: Charge and MW of peptides is not given here which makes it difficult to follow the discussion, also a categorization in groups is done, These groups e.g. AP1, AP2, AP3 should also be clear from the table. Where is the p-vale column, you mention stat tests in the subscript? It is suggested to make a e.g. graphical representation of groups and their transport (heat map?) to further clarify sections 3.3.1 and 3.3.2 (if you think this helps) or otherwise represent the data more in a more structured way. Please clarify the relation between total protein content, relative abundance and the impact of this on the rates of RA that are used to compare the protein migration among membranes. If focus is only on selectivity and not on absolute transport rates it should be mentioned/explained more clearly.

Minor remarks:

L66: water transport due to electromigration and hydrodynamic delta P, osmosis

L82: specify the method or reference

Table 1:

why is there a column “peak” what does it mean/contribute.

Peptide source “BLG” clarify abbreviation.

Molecular mass, does it need so many digits after comma?

Gravy and molecular mass: what is the measurement error?   

L117: Gravy should be explained in materials and methods not as a table subscript

If the table is taken from [15] (publication by same authors) it should be stated clearly.

L232: why not showing units for migration rate while showing units for other variables in Eq1. 

L229: effective surface of FM: what do you mean, total surface or on one side.

L 232: explain why you are not using SI units or otherwise use SI units

L318: no difference for 100-300 kDa membranes => check, Fig 3 seems to indicate differently?

L352: did rate of AP1 increase or not?

L515: Is there a reason to exclude diffusion, osmostic/electrosmotic flow, hydrodynamics etc to explain the transport of these neutral peptides? 

Conclusion could be improved (focus).

To check:

L125: each one…(reformulate)

L148: in every…(reformulate)

L67: water is a solvent

L68: selectivity performances…(reformulate)

L81: made up …(reformulate)

L286: clarify/rewrite

Author Response

The article covers the comparison of different UF membranes for use in separation of a peptide mixture with EDFM technology. The work is very interesting, the experimental approach is of high quality and the paper is generally well written. Some improvements are required however.

We would like to thank the reviewer for her/his suggestions and valuable insights on the manuscript. We have addressed each of the points raised. The manuscript was revised and updated according to the suggestions.

General remarks:

Statistical tests should be reported in scientific format to avoid any doubt. Please check:

Table 2: Anova tests (degrees of freedom, F-statistic and p-value) e.g. F(1, 24) = 44.4, p < .001.

Thanks for the pertinent comment. However, in order to focus on the data and to not overweight the tables we decided to mention some information in the legend of the tables as well as the statistical analyses part, as generally done in articles of the research field, instead of adding other columns with the F-statistics and degree of freedom for all the properties or parameter studied.

Table 2,3,4: Indicate somewhere what x ± y refers to,  is it mean and SD (not SEM? N?)

Indicated in the text as suggested. (L257-258)

Table 2: Clarify what the column P-value refers to.

Clarified (L266-267)

L245: Explain very briefly (1-2 sentences) what these stat tests do (null hypothesis, assumptions/levene’s test, what result looks like, how result is interpreted) to improve readability, as you use them a lot in the results section.

Clarified as requested (L254-257).

L250: significance level, alpha = 0.05.

Updated in the text.

L317: “ the global rate of peptide migration to A-RC was significantly lower than to C+RC (P<0.05) “ which test is performed here?

t-test was carried out for each membrane to see if the global rate of peptides migration to anionic and cationic recovery compartments were significantly different or not. This information was added in the statistical analyses part.

Section 3.3: when quantitatively comparing RA slopes of different peptides groups (AP1-AP3, CP1-CP3) among each other, ancova or regression analysis should be used. Otherwise it should be mentioned specifically on which basis qualitative comparison is made.

The RA of each peptide among tested FMs were compared by using one-way ANOVA in this study. The grouping of peptides was done later based on the trends of their abundance as MWCO of FMs increases.

Section 3.1 should be improved, not all information required to follow the story is presented in a clear way.

Improved as requested.

Table 3 and 4: Charge and MW of peptides is not given here which makes it difficult to follow the discussion, also a categorization in groups is done, These groups e.g. AP1, AP2, AP3 should also be clear from the table.

Charge and MW of peptides are presented in Table 1 as well as in the text while discussing the results of table 3 and 4. All groups of peptides (AP1, AP2, AP3…) were orderly arranged in the table.

Where is the p-vale column, you mention stat tests in the subscript?

The significant differences in RA of a peptide for different membranes were declared at probability level P<0.05. This is mentioned at the vicinity of the table.

It is suggested to make a e.g. graphical representation of groups and their transport (heat map?) to further clarify sections 3.3.1 and 3.3.2 (if you think this helps) or otherwise represent the data more in a more structured way.

RA of peptides were presented in a more structured way.

Please clarify the relation between total protein content, relative abundance and the impact of this on the rates of RA that are used to compare the protein migration among membranes. If focus is only on selectivity and not on absolute transport rates it should be mentioned/explained more clearly.

The focus of this study was on understanding the impact of physicochemical properties of PES membrane on the global migration and selective separation of peptides to recovery compartments. This information is stated in the last part of introduction section.

Minor remarks:

L66: water transport due to electromigration and hydrodynamic delta P, osmosis

During EDFM, in general, there is negligible amount of water transport through FMs. Only the charged peptides migrates through FMs. Futhermore no transmembrane pressure is applied : The migration of peptides is mainly due to the electric field.

L82: specify the method or reference

The method is specified in the text. (L84)

Table 1:

why is there a column “peak” what does it mean/contribute.

“Peak” column has been deleted from all the related tables to clarify the tables.

Peptide source “BLG” clarify abbreviation.

The abbreviation “BLG” has been clarified just below the table (L122)

Molecular mass, does it need so many digits after comma?

Number of digits after decimal have been reduced from 4 to 2 for molecular mass of peptides in the table 1

Gravy and molecular mass: what is the measurement error?   

GRAVY score was calculated by using the ExPASy Molecular Biology Server. Molecular mass of peptides was obtained by qTOF (UPLC-MS/MS-QTOF). There was no deviation in molecular mass up to two digits after decimal.

L117: Gravy should be explained in materials and methods not as a table subscript

If the table is taken from [15] (publication by same authors) it should be stated clearly.

GRAVY score was not calculated during this study, but it was taken from the previous studies. Therefore reference [15] is added in the table

L232: why not showing units for migration rate while showing units for other variables in Eq1. 

Unit for migration rate is updated

L229: effective surface of FM: what do you mean, total surface or on one side.

Effective surface means total membrane surface on filtrating side through with migration of peptides to recovery compartments take place. The effective surface of FM in this study was 100 cm2 (L126-127).

L 232: explain why you are not using SI units or otherwise use SI units

We have used g/m2.h as the unit of migration rate and not the SI units. If we use SI units, the value of migration rate will be too low with many digits after decimal. For instance, the global peptides migration rate to A-RC for 100 kDa would be 0.00000153 kg/m2.s (5.50 g/m2.h).

L318: no difference for 100-300 kDa membranes => check, Fig 3 seems to indicate differently?

For 100 kDa, global peptides migration to anionic recovery compartment was 5.50±0.70g/m2.h while for cationic recovery compartment was 7.63±1.37g/m2.h. These two values were not statistically significant (P=0.074). Similarly, for 300 kDa, global peptides migration to A-RC was 11.28±1.40g/m2.h and to C+RC was 10.92±1.65g/m2.h, which were not statistically significant (P=0.786).

L352: did rate of AP1 increase or not?

L452 . The migration rate of AP1 group of peptides increased with an increase in MWCO of PES membranes. However, their relative abundance decreased because the increase in migration rate of AP2 group of peptides was much more higher comparing to the increase in migration rate of AP1 group of peptides as MWCO increases. This is because AP2 group of peptides were highly negatively charged, while AP1 group of peptides were slightly negatively charged. Indeed, charge of the peptide plays an important role in their migration through charged membranes under the electric field. Furthermore, with an increase in MWCO, the effect of steric hindrance /size exclusion effect on AP2 group of peptides also decreases facilitating their migration.

L515: Is there a reason to exclude diffusion, osmostic/electrosmotic flow, hydrodynamics etc to explain the transport of these neutral peptides? 

Actually, the main explanation for neutral peptides migration to the recovery compartments was via hydrophobic interaction with charged peptides as mentioned in the text. However, the diffusion of peptides has never been observed in all the previous studies. May be electroosmotic flow but no information is available in the literature. Mentioned in the text.

Conclusion could be improved (focus).

Modified as requested

To check:

L125: each one…(reformulate)

Reformulated

L148: in every…(reformulate)

Reformulated

L67: water is a solvent

Solvent is replaced by chemical solvent

L68: selectivity performances…(reformulate)

Reformulated

L81: made up …(reformulate)

Reformulated

L286: clarify/rewrite

Clarified. (L295-296)

Reviewer 2 Report

The current study uses filtration membranes (FMs) as the separating membranes in electrodialysis for the fractionation of bioactive peptide.  The influence of the various membrane physicochemical properties on the separation performance was systemically studied. The experiments were well designed, and the manuscript was well organized. However, some concerns need to be resolved before the manuscript can be published. A revision is thus suggested.

The detailed comments are listed below:

The English language should be carefully checked. How does the new method proposed in this study (using FMs for bioactive peptide fractionation in electrodialysis) compared with the existing methods/technologies? Please discuss that in the manuscript. Table 2, what do the subscripts (a, b, ab) means? Lines 143-145, what is the pH value in the system? Lines 273-279, the authors discussed the change of membrane electrical conductivity with the change of the membrane MWCO. However, ionic conductivity matters but not electrical conductivity in the electrodialysis system, since the ions are transporting inside the membrane system but not the electron. Lines 498-500, the authors claimed that the negatively charged peptides aggregate with cationic peptides, and that explains why the negatively charged peptides also migrated through the membranes to C+RC. Will the same happen in the A-RC and how can that influence the selectivity? Line 515-517. The RA of the neutral peptides followed a similar trend as that of the AP1 group of the peptides in A-RC; however, the RA of the neutral peptides kept unchanged in C+RC. Why does the transport of the neutral peptides so different in the different compartments? How can this influence the system's performance? Please give some discussion.

Author Response

Reviewer 2

The current study uses filtration membranes (FMs) as the separating membranes in electrodialysis for the fractionation of bioactive peptide.  The influence of the various membrane physicochemical properties on the separation performance was systemically studied. The experiments were well designed, and the manuscript was well organized. However, some concerns need to be resolved before the manuscript can be published. A revision is thus suggested.

The detailed comments are listed below:

The English language should be carefully checked.

English language was carefully checked

How does the new method proposed in this study (using FMs for bioactive peptide fractionation in electrodialysis) compared with the existing methods/technologies? Please discuss that in the manuscript.

This new method has been compared briefly with the existing pressure driven method in the introduction section (L64-75)

Table 2, what do the subscripts (a, b, ab) means?

The subscripts are explained in the vicinity of the table

Lines 143-145, what is the pH value in the system?

The pH value of the system was 7. It is already written in the text (L148)

Lines 273-279, the authors discussed the change of membrane electrical conductivity with the change of the membrane MWCO. However, ionic conductivity matters but not electrical conductivity in the electrodialysis system, since the ions are transporting inside the membrane system but not the electron.

Since EDFM is an electro membrane process, peptides migrate through FM under the effect of electric field. Consequently, the ability of FM to conduct the electricity also matters. With an increase in FM electrical conductivity, the resistance of the system is decreased and so is the energy consumption of the process. Ionic conductivity also matters since it can interfere on the migration of peptides. Therefore, ionic conductivity of 3000mS/cm was maintained through the EDFM experiment.

 Lines 498-500, the authors claimed that the negatively charged peptides aggregate with cationic peptides, and that explains why the negatively charged peptides also migrated through the membranes to C+RC. Will the same happen in the A-RC and how can that influence the selectivity?

No, the same phenomenon did not happen in A-RC, because only negatively charged peptides were noticed in this compartment. For the peptide or peptide aggregate to be migrated to A-RC, the net charge should be negative. This could be possible only when the cationic peptides aggregates with AP2 group of peptides (having high molecular mass and high negative charge). Consequently, such a large aggregate must have been rejected by tested FM due to size exclusion effect.

Line 515-517. The RA of the neutral peptides followed a similar trend as that of the AP1 group of the peptides in A-RC; however, the RA of the neutral peptides kept unchanged in C+RC. Why does the transport of the neutral peptides so different in the different compartments? How can this influence the system's performance? Please give some discussion.

In A-RC, their RA for different FMs followed the similar trend as that of AP1 group of peptides: their RA decreased with an increase in MWCO. However, this doesn’t mean that their migration rate decreased with as increase in MWCO of FMs. As explained earlier for AP1 group of peptides, the lower RA of neutral peptides was due to high and competitive migration rate of AP2 group of peptides (due to their high charge) as MWCO of FMs increases. On the other hand, their RA in C+RC was comparable whatever the MWCO was used. This is because all the cationic peptides present in the hydrolysate in this study were small (MW<1200 Da) and slightly positively charged, and therefore, their migration rate through any FMs were not affected by steric hindrance/electrostatic repulsion. This resulted in more or less similar RA of a specific cationic peptide for all the FMs and consequently the comparable RA of neutral peptides for any MWCO tested.

This information has been added in the text. (L523-532)

Round 2

Reviewer 1 Report

The manuscript has been improved according to the suggestions.

Some minor textual details need to be optimized:

Molar masses in text still have 4 digits after comma (e.g. L473, L520, L521)

degree sign missing in L 200: “if higher than 90” °

In subscript of tables 2, 3 and 4 it is mentioned n=6. Commonly n refers to number of observations, here it seems to be indicating number of groups which is a bit unusual.

Author Response

The manuscript has been improved according to the suggestions.

Some minor textual details need to be optimized:

Minor details were optimized as suggested.

Molar masses in text still have 4 digits after comma (e.g. L473, L520, L521)

Molecular mass of peptides in the texts and table are reduced to 2 digits after comma.

degree sign missing in L 200: “if higher than 90” °

L281-282: ° sign was added.

In subscript of tables 2, 3 and 4 it is mentioned n=6. Commonly n refers to number of observations, here it seems to be indicating number of groups which is a bit unusual.

Thank you very much for the pertinent comment. n=6 has been removed from all the tables

Reviewer 2 Report

All my comments have been addressed. I think the quality of the paper has been improved, so I recommend acceptance.

Author Response

All my comments have been addressed. I think the quality of the paper has been improved, so I recommend acceptance.

Thanks a lot.
